# Comprehensive Review of Metastatic Breast Carcinoma in Cytology Specimens

Swikrity U. Baskota [1], Daniel Qazi [2], Ashish Chandra [3] and Poonam Vohra [2,*]

1    Department of Pathology and Cell Biology, Columbia University Medical Center, 630 W 168th St, New York, NY 10032, USA
2    Department of Pathology and Laboratory Medicine, University of California San Francisco, San Francisco, CA 94110, USA
3    Department of Histopathology, King's College Hospital NHS Foundation Trust, London SE1 7EH, UK
*    Correspondence: poonam.vohra@ucsf.edu

**Abstract:** Breast carcinomas are known to metastasize to various organs of the human body. Fine needle aspiration cytology or exfoliative cytology often are the standard method for diagnosis at these metastatic sites due to ease of procurement of diagnostic material, accessibility, less complications, high sensitivity, and specificity of diagnosis and evaluation of biomarker status needed to guide future management. This comprehensive review article discusses in detail metastatic patterns, cytomorphology of metastatic breast cancer at different body sites, immunohistochemistry needed for diagnosis of breast carcinoma, sensitivity and specificity of diagnosis and breast biomarker assays in the cytology material.

**Keywords:** metastatic breast carcinoma; fine-needle aspiration cytology; hormone receptors; breast biomarkers; immunohistochemistry



## 1. Introduction

Carcinoma of the female breast is the leading cause of global cancer as of 2020, accounting for 11.7% of all cancers and reported to be the fifth leading cause of cancer mortality worldwide with 685,000 deaths, and the majority of breast cancer mortality is attributed to metastatic disease [1]. The 5-year survival rate of metastatic breast cancer (MBC) is reported to be 29% [2]. Breast carcinomas are known to metastasize to almost every organ of the human body. The most common sites of metastases include bone (65.1%), lung (35.4%), liver (26.0%) and brain (8.8%) [3]. MBC has been reported in other body sites including thyroid [4,5], urinary bladder [6], ovary [7], gastrointestinal organs, gynecologic organs, peritoneum, retroperitoneum, adrenal glands, and bone marrow [8].

Fine needle aspiration cytology (FNAC) or exfoliative cytology often are the standard method for diagnosis at these metastatic sites due to ease of procurement of diagnostic material, accessibility, less complications, high sensitivity, and specificity of diagnosis and evaluation of biomarker status needed to guide future management. This comprehensive review article discusses in detail metastatic breast carcinoma patterns, cytomorphology of metastatic breast carcinoma at different body sites, immunohistochemistry (IHC) needed for diagnosis of breast carcinoma, sensitivity and specificity of diagnosis and breast biomarkers performed on the cytology material.

## 2. Metastatic Breast Carcinoma Patterns

The relationship between different molecular signatures of breast carcinoma and distant metastasis has been studied. Hormone receptor positive tumors are more likely to demonstrate bony metastasis [9]. Hormone receptor negative/ HER-2 negative tumors and hormone receptor negative/ HER-2 positive tumors are associated with visceral metastasis

such as lung and liver [10]. Patients with bone metastases are reported to have a longer overall survival than those with visceral metastases [11].

Different breast carcinoma histologic sub-types also exhibit different patterns of loco-regional and distant metastases. In a study performed on a large cohort (2605 cases) of invasive lobular carcinoma (ILC) and invasive breast carcinoma of no special type (IBC-NST) cases, no significant difference in metastases pattern was noted among lymph-nodes, liver, and central nervous system metastases. However, IBC-NST was reported to metastasize more frequently to lungs, pleura, liver, and brain, whereas ILC was reported to involve gastrointestinal organs, gynecologic organs, peritoneum, retroperitoneum, leptomeninges, adrenal glands, and bone marrow more frequently ($p < 0.05$) [8,12].

## 3. Fine-Needle Aspiration Cytology for Metastatic Breast Carcinoma

FNAC is often used to determine regional metastasis in the pre-operative staging of primary breast cancer to triage patients for sentinel lymph node biopsy versus axillary lymph node dissection as well as to make therapeutic decisions before surgery. Axillary ultrasound combined with FNAC of lymph nodes has been reported to demonstrate an accuracy of 82.2% [13]. The addition of rapid on-site evaluation (ROSE) to axillary ultrasound-guided FNAC has been found to increase the adequacy rate of these specimens from 78% to 96% [14]. The ultrasound finding of axillary lymphadenopathy of greater than or equal to 10 mm is reported to be an independent predictor of a positive FNAC. Axillary lymph nodes with ultrasound findings of extra nodal extension are also significantly associated with positive FNAC [14]. False-negative rate of pre-operative axillary ultrasound-guided FNAC has been reported up to 31% [15], which is predominantly attributed to sampling error specifically to the small size of metastatic foci in the subsequent biopsy specimen [16]. These small metastatic foci, which are usually micrometastases (<2 mm) or isolated tumor cells (<200 cells or <0.2 mm), are shown to have very sparse prognostic implication in breast cancer patients [17]. FNAC of an axillary lymph node is most helpful when it is positive, to make a pre-operative decision of sentinel node biopsy versus lymph node dissection. However, in the cases with a negative FNAC, it is still recommended to proceed with sentinel lymph nodes biopsy [15,16].

FNAC is also useful for the diagnosis and assessment of biomarker status at distant metastatic sites. As most of the MBC to bone are osteolytic, they still remain amenable to fine-needle aspiration biopsy [18]. FNA sampling of bone metastases for breast carcinoma is a well-accepted method for diagnosis, as it avoids the decalcification process used for core needle biopsies of bone. It has been reported that the decalcification process is significantly associated with discordant or decreased expression of hormonal receptor status and HER2 FISH analysis in MBC when compared with primary tumor sites [19].

FNAC of axillary lymph nodes is a simple, minimally invasive technique that is used to improve preoperative determination of the status of the axillary lymph nodes for metastases in patients with MBC [16]. However, FNA biopsy of axillary lymph nodes to assess the response to neo-adjuvant chemotherapy in cases with MBC was found to have a high false negative rate [20].

Breast cancer is the most common cancer that metastasizes to the esophagus and the periesophageal structures. EUS-FNA has been found to be safe and effective for the evaluation of breast cancer metastases to the mediastinum and the esophagus and can also provide sufficient tissue for biomarker analysis [21].

## 4. Touch-Imprints for Regional Metastatic Breast Carcinoma

The sensitivity of detecting macro-metastasis of more than 2 mm is significantly high with the concurrent use of touch-imprint cytology during intraoperative frozen section evaluation of sentinel node biopsy [22].

## 5. Exfoliative Cytology for Metastatic Breast Carcinoma

Detection of MBC cells in CSF cytology remains the standard choice for diagnosis and follow-up of leptomeningeal spread of MBC [23]. Involvement of pleural [24], peritoneal, and pelvic cavity [25] is also established by examination of cavity fluid from these sites. MBC has also been reported in rare urine cytology [6,26], bronchoalveolar lavage cytology [27] and cervical pap cytology [28,29]. There are rare case reports of MBC including lobular carcinoma in the cervical pap smears. Although relatively a rare event, it needs to be distinguished from other, more common primary gynecologic tumors and dysplasia. In these cases, previous clinical history of breast cancer is important for accurate diagnosis [30,31].

## 6. Cytomorphology of Metastatic Breast Carcinoma in FNA Specimens

The cytomorphology of MBC is dependent on the primary histologic sub-type of the breast carcinoma cells. The metastatic carcinoma cells arising from IBC-NST are usually a mixture of tissue fragments and single cells. The examples of metastatic breast cancer to bone and thyroid are illustrated in Figures 1A–C and 2A–D, respectively. The tissue fragments are usually well circumscribed, round to oval cells with a focal acinar formation. The individual cells can demonstrate significant nuclear pleomorphism. On the contrary, the metastatic carcinoma cells arising from lobular carcinoma are predominantly single cells with rare tissue fragments and are usually monotonous in appearance with a markedly high nuclear: cytoplasmic ratio. The presence of single cells in a cytology specimen poses a diagnostic challenge and often needs to be distinguished from discohesive malignant tumors such as lymphoma and melanoma [12].

A few case reports of cytomorphologic features of metastatic mucinous carcinoma of breast to thyroid and parotid gland have been reported [4,32]. On FNA smears, the tumor cells are arranged in clusters or as single cells and demonstrate low to intermediate grade nuclei and moderately abundant eosinophilic cytoplasm with intracytoplasmic mucin. Some of the cells may show eccentric nuclei. The cell block highlights tumor cells floating in lakes of mucin. The patient's clinical history and IHC are critical in these cases to provide valuable information. In cases of metastatic mucinous carcinoma to thyroid, IHC on cell block including negative staining with TTF-1, thyroglobulin and calcitonin can help to differentiate these tumors from thyroid mucinous carcinoma, papillary carcinoma and follicular adenoma with mucus secretion, and thyroid medullary carcinoma, respectively [32].

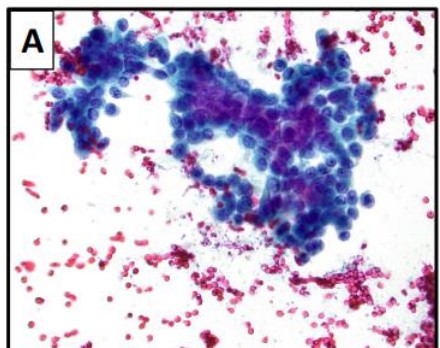 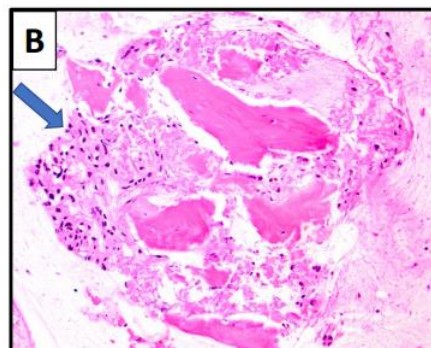 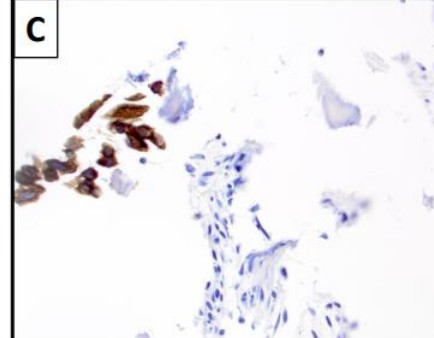

**Figure 1.** (**A**–**C**) Metastatic breast cancer in bone. (**A**) FNA smear shows clusters of loosely cohesive tumor cells with moderate pleomorphism and prominent nucleoli. (**B**) Cell block shows tumor cells (arrow) in association with bone fragments. (**C**) Tumor cells show positive staining with pancytokeratin consistent with metastatic carcinoma.

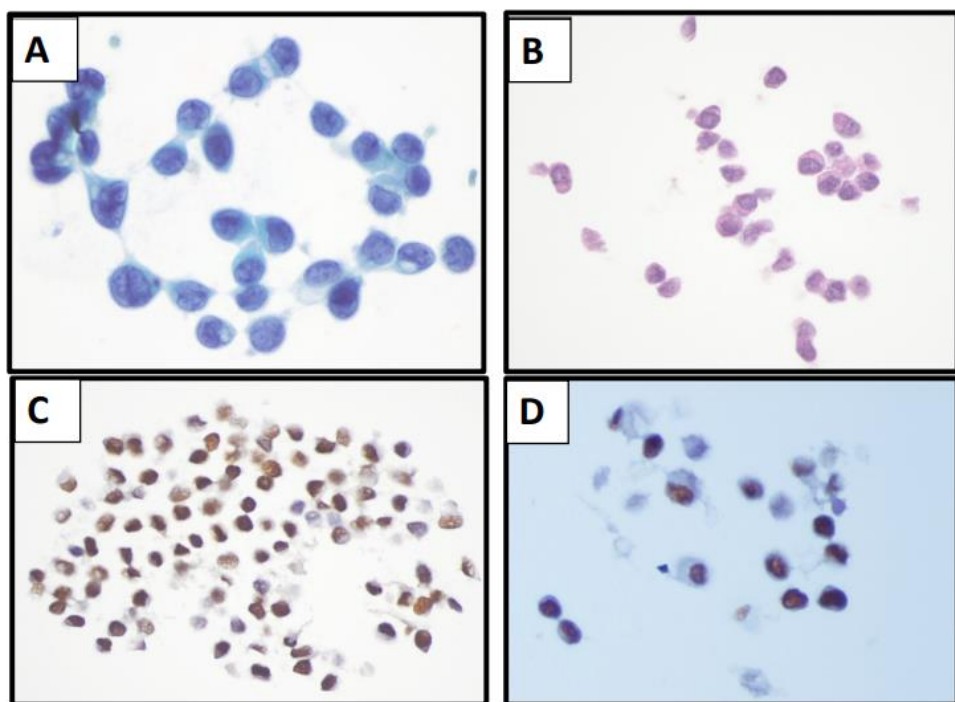

**Figure 2.** (**A**–**D**) Lobular carcinoma of breast metastatic to thyroid. (**A**) Dyscohesive single cell pattern of moderately pleomorphic tumor cells with intracytoplasmic lumina (Pap Stain). (**B**) Cell block is concordant and highlights single cell pattern (H&E stain). (**C**) Immunohistochemical stain for ER shows strong positive staining in the majority of tumor cells. (**D**) GATA-3 shows positive staining in tumor cells consistent with breast primary (Images contributed by John P. Crapanzano, MD, Columbia University Medical Center).

The cytomorphologic findings of metastatic micropapillary carcinoma of breast, a rare variant of IBC, on FNAC of lymph nodes, has been described in rare case reports [33,34]. The cytomorphologic features include presence of oval to angulated, three-dimensional clusters of tumor cells/tumor morules arranged in papillary to tubuloalveolar architecture without definite evidence of fibrovascular cores. Focally, the clusters/morules of the tumor cells are separated by small, slit-like spaces. The architecture of these clusters in cytology is similar to that seen in histologic sections of these tumors. The tumor cells demonstrate mild to intermediate nuclear-grade atypia with irregular nuclear contours and finely dispersed chromatin. Focal areas of discohesive/isolated single tumor cells with significant nuclear atypia can be present [33,34]. The "reverse polarity" highlighted by IHC for epithelial membrane antigen can be performed on cell block material to further support the diagnosis [33,34].

Metaplastic carcinoma (MC) of breast constitutes a heterogenous group of tumors with distinctive morphology but with marked intertumoral and intratumoral heterogeneity. Due to the heterogeneous nature of this tumor, it is not uncommon for only one component of MC to be identified at a metastatic site, making cytologic interpretation of metastatic MC on FNAC challenging. Presence of chondromyxoid stroma, other mesenchymal elements, bland spindle cells, markedly pleomorphic tumor cells including atypical squamous cells should raise the possibility of metastatic MC in the right clinical context of prior history of MC of breast [5,35,36].

## 7. Cytomorphology of Metastatic Breast Carcinoma in Effusion Specimens

The cytomorphology of MBC is variable in effusion specimen and can demonstrate non-cohesive isolated cells, cohesive cell groups, linear arrangements, and large cell balls [37,38]. Some cases show high-grade nuclear features, whereas others have mild nuclear atypia and can mimic reactive mesothelial cells. New effusions with IBC-NST

usually show isolated or small loose clusters of small- to medium-sized cells with hyperchromatic, round to oval nuclei, small inconspicuous nucleoli, and scant cytoplasm with a large, single cytoplasmic vacuole in some cells. A "cell-within-a-cell" arrangement is commonly noted. Some of the neoplastic cells wrap around another carcinoma cell giving an appearance of small epithelial pearls. In these cases, single cells arrangement, polygonal cell shape and fine chromatin can be helpful in distinguishing another common metastatic ovarian adenocarcinoma cells, which are often seen in three-dimensional cell groups and exhibit coarse chromatin [39]. The isolated cell pattern or small loose clusters of adenocarcinoma cells in effusion samples can mimic and can be challenging to distinguish from reactive mesothelial cells and histiocytes (Figure 3A). Identifying a discrete population of atypical cells distinct from mesothelial cells is the key for diagnosing malignant effusions. In challenging cases where cytomorphological features of adenocarcinoma and reactive mesothelial cells are overlapping, cell block preparation in conjunction with IHC staining is critical for an accurate diagnosis [40] (Figure 3B–D).

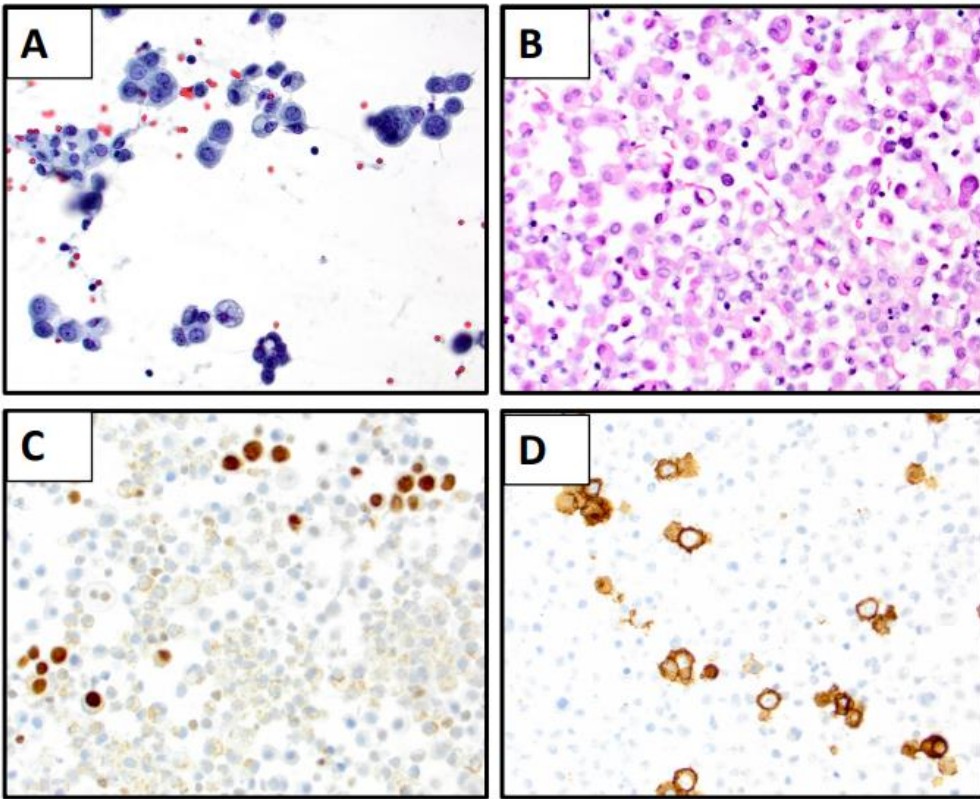

**Figure 3.** (**A**–**D**) Metastatic breast cancer in pleural fluid. (**A**) Smear demonstrates predominantly isolated cell pattern—can be a diagnostic pitfall/difficult to differentiate from reactive mesothelial cells and histiocytes. (**B**) Cell block can be helpful to perform IHC, as reactive mesothelial cells can mimic tumor cells. (**C**) GATA-3 highlights tumor cells with negative staining in mesothelial cells and histiocytes. (**D**) HER2 showing strong membranous staining in tumor cells.

Recurrent or long-standing effusions in MBC may show more cohesive cell pattern with characteristic three-dimensional proliferation spheres of various sizes. These three-dimensional proliferation spheres are composed of cells with scant and non-vacuolated cytoplasm. The nucleiare arranged in longitudinal arrangement along the periphery of the spheres. Occasionally, conglomerations of proliferation spheres may lead to papillary-like configurations. Large cell balls with smooth community borders, also known as "cannon balls", are suggestive for MBC (Figure 4A–D). Abundant hollow spheres are a common pattern in MBC seen in cell block preparations [41] (Figure 4E). Immunohistochemical stains such as GATA3 can be performed on a cell block to highlight tumor cells to confirm

the diagnosis (Figure 4F). The carcinoma cells in large cell patterns seen in metastatic poorly differentiated IBC-NST tend to be scattered singly or as loosely cohesive groups with hyperchromatic nuclei, prominent nucleoli and non-vacuolated cytoplasm. Rare proliferation spheres and no papillary configurations are noted in these cases [37].

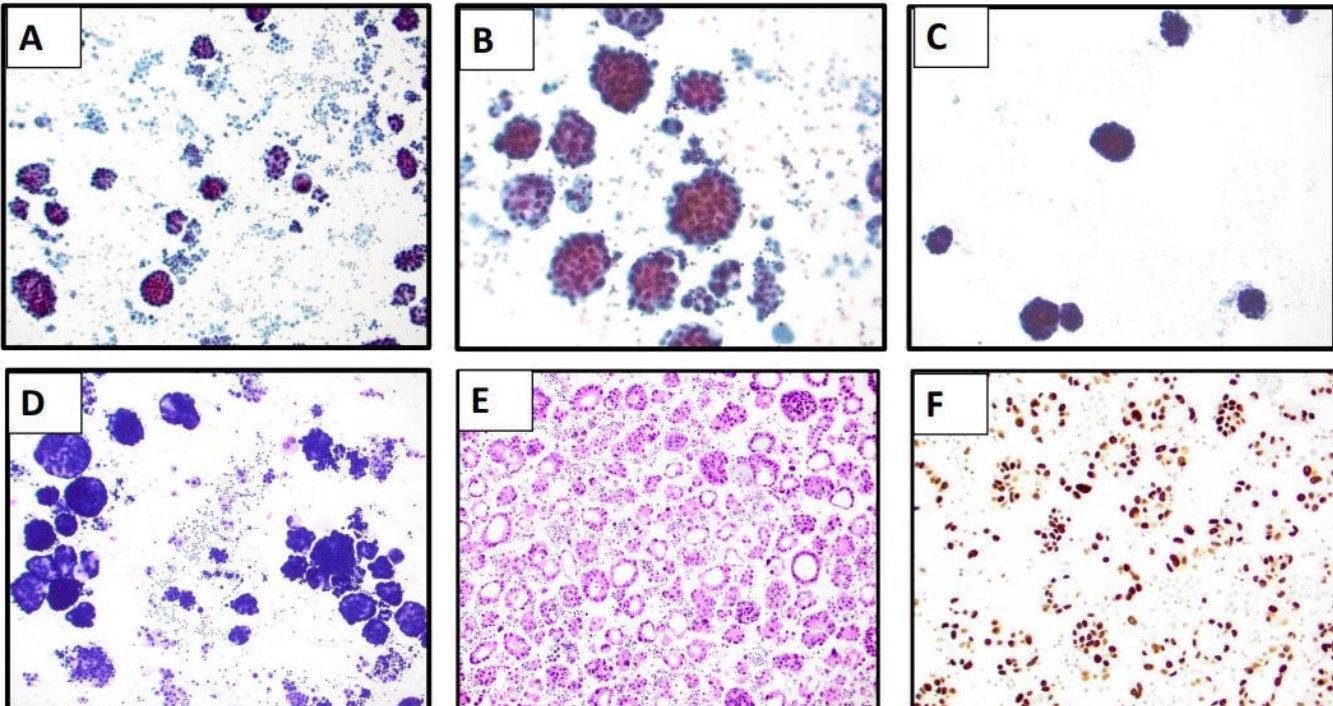

**Figure 4.** (**A–F**) Metastatic breast cancer in pleural fluid. (**A,B**) Malignant cells exfoliate as large spheres with smooth community borders in smears (Papanicolaou stain). Background shows reactive mesothelial cells and histiocytes. (**C**) ThinPrep (Papanicolaou stain). (**D**) MGG stain. (**E**) Cell block highlights hollow spheres of tumor cells, a common pattern seen in MBC. (**F**) GATA-3 shows positive staining in tumor cells, whereas mesothelial cells and histiocytes present in background are negative.

It has been reported that malignant cell spheroids in pleural fluid are associated with a better prognosis when compared to patients presenting with an isolated cell pattern with a significant difference between these groups ($p < 0.05$) [42,43].

Lobular carcinoma of the breast is subtle among all the adenocarcinomas in pleural effusion specimens, and it is extremely difficult to distinguish from histiocytes and mesothelial cells. The single cell/mesothelial-like cell pattern of lobular carcinoma demonstrates monotonous cells with eccentric nuclei, cytoplasmic vacuoles, and intracytoplasmic lumens [44]. Awareness of these cytomorphologic patterns, special stains for mucin (mucicarmine and PAS-D), IHC panels that include markers for epithelial and mesothelial cells and breast biomarkers on any effusion from a patient with lobular breast cancer can be very helpful to make a definitive diagnosis [41].

Pleomorphic lobular carcinoma (PLC) of breast, a subtype of ILC, has more aggressive clinical behavior and high-grade cytology comprising large cells with pleomorphic nuclei, prominent nucleoli and cytoplasmic blebs (Figure 5A,B). Metastatic PLC in effusion specimens can be confused with reactive/atypical mesothelial cells, as they share similar cytomorphological features. Cell block preparation (Figure 5C), the IHC staining pattern of PLC, and the review of patient's prior breast biopsy (Figure 5D) can help to distinguish it from reactive/atypical mesothelial cells and other metastatic neoplasms. However, PLC can show apocrine features, which can cause nonspecific IHC positivity that may lead to diagnostic pitfall. For instance, immunostaining with E-cadherin can show granular cytoplasmic staining in apocrine cells leading to a false positive staining pattern and therefore metastatic PLC could be misinterpreted as IBC with apocrine features [45]. Similarly, it

has been reported that 25% of apocrine breast carcinomas can show positive staining with calretinin [46]. This can make the distinction between reactive mesothelial cells and PLC challenging in serous effusion specimens [46].

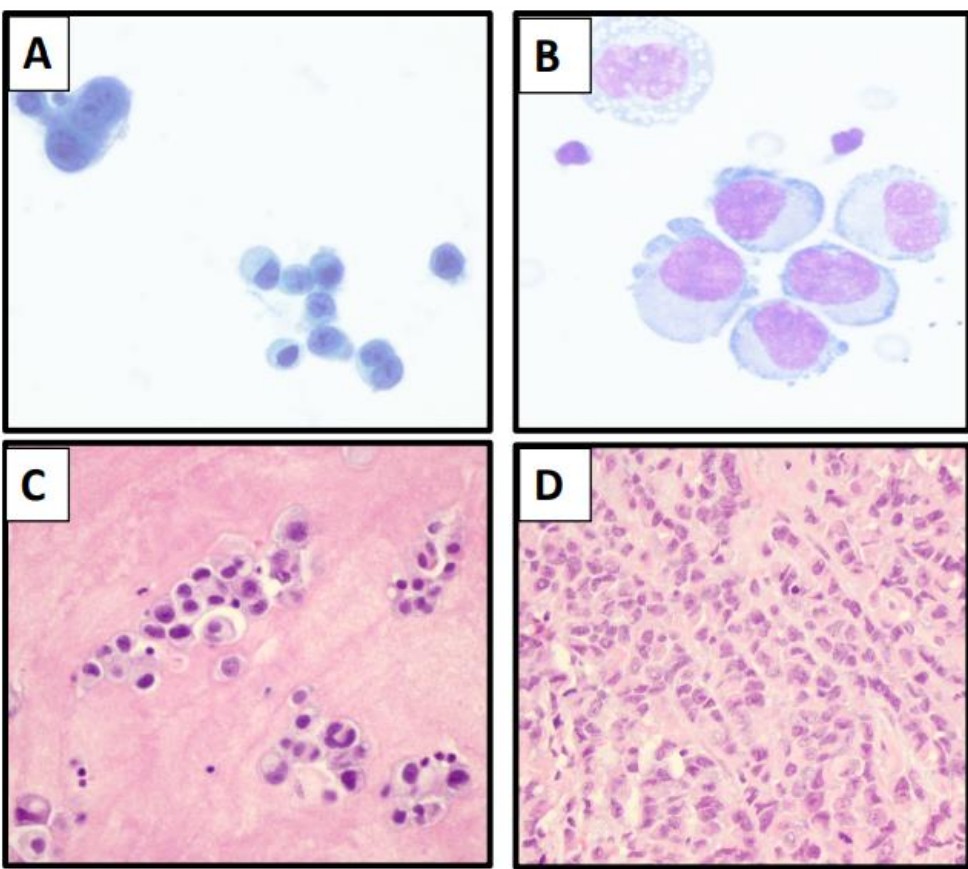

**Figure 5.** (**A–D**) Metastatic pleomorphic lobular carcinoma in pleural fluid. (**A,B**) Clusters of tumor cells with large cell size, pleomorphism and prominent nucleoli and cytoplasmic blebs. (**A**) ThinPrep (Papanicolaou stain) (**B**) DQ stain. (**C**) Cell block demonstrates small clusters and single pleomorphic cells. Cells have moderately abundant cytoplasm. (**D**) Concurrent breast biopsy with sheets of pleomorphic lobular cells cytomorphologically similar to the cell block specimen. (Images contributed by John P. Crapanzano, MD, Columbia University Medical Center).

## 8. Cytomorphology of Metastatic Breast Carcinoma in CSF Specimens

MBC is one of the most common metastatic carcinoma seen in CSF specimens [23]. MBC cells in cerebrospinal fluid (CSF) specimen tend to exhibit single cells or rarely loose clusters with moderate pleomorphism, increased nuclear:cytoplasmic ratio (mean 0.70), round to oval nuclei, finely granular chromatin, single or multiple prominent nucleoli, granular cytoplasm with distinct cell borders [47] (Figure 6A). Some cells can be binucleated. The large cannonball-like arrangements typical of MBC in pleural fluid are almost never seen in CSF. Cytoplasmic blebs can be seen in MBC in CSF specimens (Figure 6B); however, this is not a specific cytologic finding and they have been reported in metastatic carcinomas from other sites, including lung and ovary [48]. Some lobular breast cancer cells in CSF form linear arrangements as seen in cases with small cell carcinoma. Rare cases can show tumor cells with prominent intracytoplasmic granules in CSF smears in MBC with acinar differentiation [49].

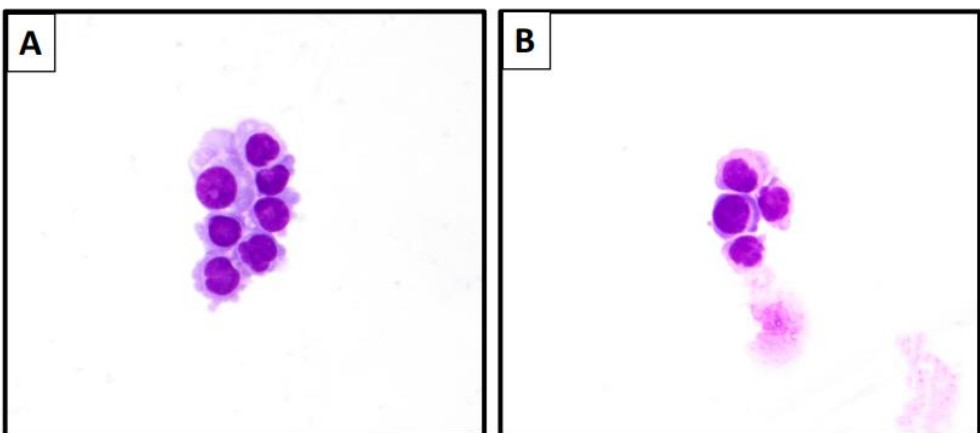

**Figure 6.** (**A**,**B**) Metastatic breast cancer in CSF. Malignant cells are large and variable in size. Nuclei are round to irregular with prominent nucleoli and moderately abundant cytoplasm with cytoplasmic blebs (Papanicolaou stain).

See Table 1 for detailed summary of cytomorphology of MBCs in different specimen types.

**Table 1.** Cytomorphology of MBC in various cytology specimen types (summarized from the references [12,23,32–43]).

| Specimen Type | Characteristic Cytomorphologic Features |
|---|---|
| Fine Needle Aspiration | Invasive breast carcinoma of no special type<br>Well-circumscribed tissue fragments with round to oval cells with focal acinar formation<br>Lobular Carcinoma<br>Single cells or small loose clusters or single file arrangement<br>Monotonous nuclei with high N:C ratio<br>Scant to moderate cytoplasm<br>Intracytoplasmic lumens/vacuoles |
| Effusion Specimens | Invasive breast carcinoma of no special type<br>Three dimensional tightly packed groups (cannonball-like clusters)<br>Hollow clusters on cell block preparation<br>Lobular Carcinoma<br>Single cell, signet ring cells<br>Mesothelial-like cell pattern<br>Single files<br>Mildly atypical cells<br>Moderately abundant cytoplasm |
| Cerebrospinal Fluid (CSF) | Invasive breast carcinoma of no special type<br>Single cells, rarely cannon ball-like arrangements<br>Large cells, prominent nucleolus<br>Occasional binucleation<br>Highly variable in size<br>Cytoplasmic blebs<br>Lobular Carcinoma<br>Isolated, medium sized cells<br>Signet ring morphology |

## 9. Immunocytochemistry for the Diagnosis of MBC

GATA-3 is one of the more sensitive markers currently widely adopted in the diagnosis of MBC including serous effusion specimens [50]. It has been reported to be more sensitive and specific than mammaglobin and GCDFP-15 for the diagnosis of MBC [51]. However, GATA-3 is not specific by itself for the diagnosis of MBC, as it can be positive

in other tumor types such as urothelial carcinoma, squamous cell carcinoma, and ovarian serous carcinoma [52]. In our clinical practice, irrespective of the specimen type and sites, a strong GATA-3 positivity with the positivity of at least one other breast marker (GCDFP-15, mammaglobin) or breast biomarkers (ER, PR and HER-2) is sufficient to diagnose MBC. However, rare entities such as the micropapillary variant of metastatic urothelial carcinoma, which reportedly expresses strong GATA-3 and frequently shows HER-2 expression, should also be considered in the differential diagnosis. Positive staining for cytokeratin-20, uroplakin-3, GATA-3 and HER-2 favors a urothelial primary over breast primary [53].

The newer markers such as trichorhinophalangeal syndrome-1 (TRPS-1) [54] appear promising in the diagnosis of challenging MBC subtypes such as triple-negative breast carcinoma, metastatic lobular breast carcinoma, metastatic invasive breast carcinoma, and metaplastic breast carcinoma with a very high sensitivity of 95–100%. The triple-negative apocrine carcinoma is the only subtype of triple-negative breast carcinoma, which exhibits positive GATA-3 and negative TRPS-1. TRPS-1, SOX-10 and GATA-3 are the three recommended IHC to diagnose metastatic triple-negative breast carcinoma. The sensitivity of GCDFP-15 and mammaglobin have been reported up to 18% and 23%, respectively, for the diagnosis of metastatic triple-negative breast carcinoma [55]. The role of TRPS-1 in the diagnosis of MBC in cytology specimens is still being studied; thus far, the results seem promising, specifically in the diagnosis of metastatic triple-negative breast carcinoma [56] and different specimen subtypes such as effusion cytology [57].

Workup of a metastatic breast carcinoma with known or unknown primary should include breast-specific markers such as GATA-3, mammaglobin, GCDFP-15, ER, PR, and HER-2. The panels should be expanded to include other common metastatic carcinomas such as TTF-1, cytokeratin-7 (for lung), cytokeratin-20 (urothelial, colon), PAX-8 (gynecological and ovarian) and CDX2 and SATB2 (gastrointestinal). In cases with rare histologic subtypes of breast carcinoma, triple-negative breast carcinoma or post-therapy with clinical suspicion of receptor conversion, the panel should be expanded to include more markers such as TRPS-1, high-molecular weight cytokeratin and SOX-10. Evaluation of cytomorphology, extended immunohistochemical panel, along with integration of clinical history and imaging information might be helpful in suggesting primary tumor type in challenging cases.

Flow cytometry (FC) immunophenotyping can be utilized to differentiate adenocarcinoma from reactive mesothelial cells and reported to have similar sensitivity and specificity comparable to IHC. However, its use is limited to diagnostically challenging cases where morphology and IHC are equivocal for the diagnosis of malignant effusion [58].

## 10. Biomarker Analysis in Cytology Specimens

In breast carcinoma, biomarker conversion at metastatic sites from the primary breast carcinoma is reported in up to 16% for estrogen receptors (ER) and 10% for human epidermal growth factor receptors-2 (HER-2), which necessitates the revision of therapeutic intervention [59,60]. Current widely adopted guidelines recommend testing for biomarkers at every new MBC site [61,62]. ASCO/CAP guidelines recommend the optimal cold ischemic interval (time between the removal of sample from the patient to the fixing of the specimen in 10% formalin) for biomarker analysis to be less than 1 h, including transport. It further recommends the specimen to be fixed in formalin for at least 6 h, and not to exceed 48 h for HER-2 testing and can be fixed up to 72 h for ER/PR testing [63,64]. These specific and strict guidelines for quantitative biomarker assays are not yet validated for cytology samples (Table 2). Numerous factors such as formalin or fixatives used in liquid-based collection fluid or different cytologic preparations such as smears, cytospins, ThinPrep, and cell block may affect the biomarker assay. However, many international studies examining various cytologic preparations (formalin vs CytoLyt fixation) [65] have shown strong correlation with concurrent histologic material, thus supporting the use of cytology material alone for the study of hormone receptor assays [25,66–70]. The cell blocks have been found to be particularly useful in the assessment of breast biomarkers in MBC to plan treatment,

potentially obviating the need for additional biopsies [66,70,71]. Excellent concordance has been reported between HER2 IHC performed on a cell block and HER2 FISH results, as well as between HER2 FISH performed on a cell block compared with tissue block [66,72].

**Table 2.** Key points regarding breast biomarkers [61–66].

| Key Points Regarding Breast Biomarkers |
| --- |
| <ul><li>Current guidelines recommend biomarker testing at every new metastatic breast cancer site</li><li>Optimal cold ischemic interval should be less than 1 h</li><li>Specimens should be fixed in formalin for at least 6 h, and no greater than 48 h for HER-2 testing and no greater than 72 h for ER/PR testing</li><li>Although strongly correlated with histology specimens, processing and interpretation guidelines have not been validated for cytology specimens</li></ul> |

Some of the factors that have been reported to affect the hormone receptors study in cytology specimens are fixatives (formalin: preferred) and the antigen retrieval methodology [69]. These limitations were noted more with HER2 IHC, especially in fluid specimens and in cases with scant material [73].

## 11. Molecular Analysis of MBC

Similar to the hormone receptors conversion seen in the MBC specimens, tumor heterogeneity and the mutational landscape can also differ in MBC specimens from the primary breast carcinoma sites. The archival material obtained via FNAC can also be used to reflex molecular analysis at the metastatic sites, specifically for hormone resistant, hormone receptor negative aggressive breast cancer subtypes. A recent pilot study [74] demonstrated the use of FNAC material for studying the mutational landscape, but a large FNAC specimen study of MBC is needed to further understand the molecular biology of MBCs.

## 12. Newer Techniques

In the era of precision medicine and personalized cancer treatment, significant work is still needed to improve the accuracy of diagnosis and to identify the therapeutic targets of MBC. Use of digital pathology techniques and point-of-care devices is still under investigation to assess potential use in early diagnosis and identification of therapeutic targets such as HER-2 receptors [75].

With the advent of new therapeutic agents such as Trastuzumab–Deruxtecan targetable for metastatic treatment, resistant or unresectable advanced primary breast carcinoma studied in cases with low HER-2 expression (defined as IHC scores of 1+ and 2+ but negative FISH) [76], a more in-depth and cytology-specimen-focused study of HER-2 protein expression in MBCs is needed.

## 13. Conclusions

Cytologic assessment of MBC specimens continues to be the first line for diagnosis as well as assessment of therapeutic and prognostic measures. Ease of obtaining the material, less complications, less material needed for diagnosis, high diagnostic sensitivity, specificity, and feasibility to perform biomarkers and other prognostic assays on cytologic preparations are a few advantages of evaluating MBC in cytology specimens.

**Author Contributions:** All the authors equally contributed to this article (study conceptualization; writing–original draft; writing–review and editing). All authors have read and agreed to the published version of the manuscript.

**Funding:** This research received no external funding.

**Institutional Review Board Statement:** Not applicable.

**Informed Consent Statement:** Not applicable.

**Data Availability Statement:** Not applicable.

**Conflicts of Interest:** The authors declare no conflict of interest.

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
