# Peer review of "Comprehensive Review of Metastatic Breast Carcinoma in Cytology Specimens"

_jmp, doi:10.3390/jmp3040025_

Round 1
Reviewer 1 Report
This is a timely review with practical implications.
The authors have described the morphological features of the IDC and ILC at metastatic sites in this brief review with good detail. However, it would also be interesting to the readers to understand how the work up the of the metastatic cancers using the appropriate ancillary IHC stains. In triple negative breast cancers the addition of TRPS1 to the existing panel of breast markers (Gata 3, mammaglobub, GCDFP-15) has helped confirm breast origin. In order to make this review more comprehensive and complete I suggest adding a paragraph on the IHC work up.
Reviewer 2 Report
First, I would like to thank you for inviting me to review this interesting manuscript.
The authors present a comprehensive review of metastatic breast carcinoma in cytology specimens. The manuscript is well written in terms of clarity, style, and use of English and has a logical construction. The figures are of good quality. The references are appropriate and current.
Minor issues
1) Isolated tumor cells are < 200 cells or less than 0,2mm
2) Invasive ductal carcinoma should be substituted with invasive breast carcinoma of no special type
3) The authors review the findings of invasive carcinoma of NST and invasive lobular carcinoma. Are there any specific cytological findings in other special types of carcinoma (mucinous or micropapillary type carcinoma for example)?
4) The authors mention that: “In our clinical practice, irrespective of the specimen type and sites, a strong GATA-3 positivity with the positivity of at least one other breast markers (GCDFP15, mammaglobin) or breast biomarkers (ER, PR, and HER-2) is sufficient to diagnose MBC.” I agree that this is true in almost all cases. One possible exception would be metastatic micropapillary urothelial carcinoma which is almost always GATA-3 and very often HER-2 positive.
5) The authors mention that “pleomorphic lobular carcinoma can show apocrine features which can cause nonspecific IHC positivity that may lead to diagnostic pitfall.” Please be more specific. Which antibody/antibodies?
